# Survey about the Quality of Life of Italian Patients with Fabry Disease

**DOI:** 10.3390/diseases9040072

**Published:** 2021-10-15

**Authors:** Barbara Polistena, Donato Rigante, Ludovico Luca Sicignano, Elena Verrecchia, Raffaele Manna, Daniela d’Angela, Federico Spandonaro

**Affiliations:** 1Department of Economic and Finance, University of Rome Tor Vergata, C.R.E.A. Sanità, via Columbia n. 2, 00133 Rome, Italy; daniela.d.angela@uniroma2.it (D.d.); federico.spandonaro@uniroma2.it (F.S.); 2Department of Life Sciences and Global Health, Fondazione Policlinico A. Gemelli IRCCS, 00168 Rome, Italy; 3Rare diseases and Periodic Fever Research Center, Department of Internal Medicine, Fondazione Policlinico Universitario A. Gemelli IRCCS, 00168 Rome, Italy; ludovicoluca.sicignano@policlinicogemelli.it (L.L.S.); elena.verrecchia@policlinicogemelli.it (E.V.); raffaele.manna@policlinicogemelli.it (R.M.)

**Keywords:** Fabry disease, health-related quality of life, personalized medicine, innovative biotechnologies

## Abstract

Fabry disease (FD) is a genetic disease included in the group of lysosomal storage disorders, caused by X-linked deficiency of the enzyme alpha-galactosidase A. The aim of this study was to evaluate different aspects related to the quality of life (QoL) of a multicentre cohort of Italian patients with FD. An observational survey was conducted to measure health-related quality of life (HR-QoL) in FD patients using the CAPI (Computer-Assisted Personal Interview) method: 106 patients (mostly women) responded to the questionnaire. Geographically, 53.7% of patients lived in northern Italy, 18.9% in central Italy and 27.4% in southern Italy or the Islands. All data were collected through a five-dimensional EuroQoL questionnaire referring to functional aspects (mobility, personal care, routine activities) and perception of physical/mental well-being (pain or discomfort, anxiety or depression). A descriptive analysis of responses was performed; FD patients were compared in terms of QoL with subjects suffering from other chronic diseases, such as Crohn’s disease, chronic hepatitis, cirrhosis and multiple sclerosis. Difficulty in normal daily activities was reported by 47.2% of FD patients. About one third of subjects also had mobility difficulties. Feelings of loneliness and isolation were reported by 33.3% of those being 60–69 years old. Anxiety was equally reported in both oldest and youngest patients (66.7%), while depression, relational problems, fear of other people’s judgement increased along with age, reaching 66.7% in the over-70-years group. Male patients were largely troubled about the risk of physical disability, particularly those aged 60 years or over. Furthermore, FD patients had a poorer QoL than people suffering from other chronic inflammatory disorders. Our study upholds that FD patients have a poor QoL, as already known, negatively impacting psychic well-being and social activities. Our survey has also found a worse QoL in FD patients compared with other severe chronic disorders.

## 1. Introduction

Fabry disease (FD) is a rare X-linked lysosomal storage disorder (LSD) caused by the deficiency of the enzyme alpha-galactosidase A and basically characterized by the progressive accumulation of complex glycolipids in the vascular endothelium of several organs, including kidney, heart, nervous system, and skin [1]. Therefore, the disease is marked by increasingly severe and permanent damage to different tissues and systems. In particular, non-catabolized lipids can accumulate in the kidney, where they cause microalbuminuria, proteinuria, and progressive loss of function up to renal failure, and in the myocardium, where they cause biventricular hypertrophy and arrhythmias. Furthermore, sphingolipids accumulate in the central nervous system with increased risk of ischemic strokes, and in the peripheral nervous system, typically causing sweating defects, heat intolerance, acroparesthesia, and gastrointestinal symptoms [1]. FD patients might also display ocular, audiological and skin involvement with occurrence of angiokeratomas, while male patients are usually characterized by the most severe and complete clinical phenotypes [1]. Treatment of FD is based on two main options: enzyme replacement therapy (ERT), agalsidase alfa and beta, administered intravenously every 14 days, and chaperone therapy (migalastat), taken orally every other day [1,2], which is only indicated for patients with FD caused by specific genetic variants defined as “amenable”.

Previous studies showed that FD can affect patients’ health-related quality of life (HR-QoL), leading to a severely compromised social life. In particular, neuropathic pain and anhidrosis have been shown to be predictors of a poorer quality of life (QoL) [3,4]. FD patients are also considered at higher risk of developing neuropsychiatric symptoms, such as depression or neuropsychological deficits [5,6,7,8,9]. The available medical literature shows that depressive symptoms are frequent in both males and females, ranging from 15 to 62% of cases [8]. Cole et al. reported that 46% among 296 FD patients presented signs and symptoms compatible with significant depression [10], while Wang et al. noted a reduced QoL even in heterozygous FD women, with depression observed in 62% of cases [11]. Depression does not follow the classic gender norm in FD, involving more frequently males than females [12].

The aim of this study was to assess how FD may change the perception of oneself and influence patients’ QoL through a targeted questionnaire. The study was promoted by the Anderson-Fabry Italian Association (AIAF) and supported by Amicus Therapeutics.

## 2. Patients and Methods

We performed this observational study to measure the living conditions and HR-QoL in a multicentre cohort of Italian FD patients. We collected data using specific questionnaires administered through the CAPI (Computer-Assisted Personal Interview) method. Patients or their caregivers answered questions through a web-based platform, and data were provided anonymously. The study was promoted by the AIAF, which, in October and November 2018, advertised the questionnaires through social networks.

In particular, the EuroQol five-dimensional questionnaire (EQ-5D; version envisaging 3 levels of answers) was administered. This EQ-5D is a simple, very short and user-friendly questionnaire, validated for the Italian version and also for self-administration, which can provide a measure of HR-QoL. In addition, it is validated for self-administration [13]. In the last few years, the EQ-5D has been largely used in Italy for evaluating individual health states in different conditions [13]. For this study, the Italian adaptation made by the EuroQoL Group was adopted [14]. The questionnaire consisted of two parts: the first one included five items: three referred to functional aspects (mobility, personal care, routine activities) and the other two to perception of physical and mental well-being (pain or discomfort, anxiety or depression). For each item, there were three possible answers indicating the absence or presence of moderate or severe problems. A special algorithm allowed the mapping of the EQ-5D, translating results in terms of quality-adjusted life years (QALY). The second part of the questionnaire consisted of a graduated scale from 0 to 100 (Visual Analogue Scale, VAS), on which the subject indicated his/her perceived health status.

For the elicitation of HR-QoL and its expression in QALY the algorithm proposed by Scalone et al. (2015) was applied to the Italian population [15]. The processing was replicated by using the algorithm developed by Dolan et al. (1995) for the United Kingdom population [16]. A specific section of the questionnaire was dedicated to further investigating the psychological issues caused by FD with special focuses on embarrassment about their condition, anxiety about the future evolution of the disease, depression, loneliness, isolation and plan to have children. Descriptive analysis and Chi-square tests for categorical variables have been carried out on data collected, assuming statistical significance if *p* ≤ 0.1.

## 3. Results of the Study

The survey involved 162 patients, 106 of whom (65.4%) completed the questionnaire. The respondents were aged between 5 and 77 years, with an average age of 42 years. The mean age at diagnosis was 31.6 years with a minimum of 1 year and a maximum of 66 years (Table 1). Women represented 59.5% of our sample. Geographically, 23% lived in northwest Italy, 31% in northeast Italy, 19% in central Italy and 27% in southern Italy or the Italian islands (Figure 1).

As for the schooling rate, 14.0% of the sample had a specific degree, 34.3% achieved a high school diploma, 44.4% had a middle or elementary school diploma, and only 1.7% did not have any school qualifications. The majority of FD patients (82.1%) received Fabry-specific treatment: in particular, 74.5% were on ERT, 7.6% were taking oral therapy, whereas 17.9% did not take any treatment.

Regarding the “routine activities” section of the EQ-5D questionnaire, 45.3% of patients had difficulty in carrying out normal daily activities, while 1.9% said that they were unable to do them at all. Regarding mobility, 28.3% of FD patients reported having difficulty moving, and a further 0.9% reported being bedridden (Table 2).

Moreover, 46.2% of patients reported experiencing moderate-to-severe anxiety; 22.7% of the sample reported feeling depressed, and 21.7% feeling lonely and isolated. In addition, 24.6% of patients said they fear the judgement of others, and 40.5% felt to be deeply influenced in the plan to have children (Table 3).

Analyzing these data by age and gender, difficulties were more common in patients over 50 years of age and in women. Feelings of loneliness and isolation were reported by 33.3% of patients in the 60–69 year group. Anxiety was reported equally by the over-70s and the youngest patients (66.7%), while depression, relational problems and fear of other people’s judgement increased with age, reaching 66.7% in the over-70 group. Comparative analysis of data between the two sexes showed that 50.8% of women suffered from anxiety, defined as moderate-to-severe, compared to 39.6% of males. Symptoms of depression were referred by 23.8% of females (versus 20.9% of males), while 23.8% of them complained of feelings of solitude (versus 18.6% of males). As expected, the impact of Fabry on reproductive decision-making was more frequent among younger patients, reaching 77.8% in those between 18 and 29 years.

Using the EQ-5D questionnaire, we found that 50.9% of patients felt “moderately” anxious or depressed and 6.6% “extremely” so (Table 2). Our data confirmed that problems were more relevant for women. (Table 3).

## 4. Disease-Related Problems

Fatigue is largely referred by FD patients with different percentages between males and females; in particular, we found a higher frequency in females (66.7% versus 58.1%, respectively) (*p* value < 0.05). With specific reference to pain-related problems, there were no significant gender differences (67.5% for males versus 69.8% for females); however, chronic/persistent pain mainly affected subjects over 70 years and 47.2% of patients lived with acute/temporary pain and 39.6% with chronic/persistent pain (Table 3).

The answers to the EQ5D questionnaire confirmed the impact of pain on FD patients: 68.9% reported experiencing pain and discomfort, and 8.5% reported experiencing it intensely (Table 2). Regarding acceptance of the disease, the questionnaire showed that 42.5% of patients accepted the disease and said they tried to live the same life as before; 36.8% accepted it and looked for the best condition to live with it; 17.9% reported they had mood swinging between acceptance and rejection, while 1.9% accepted it, but failed to react. Finally, 0.9% of patients did not succeed in “accepting” the disease.

## 5. Concerns about the Future

We found that 83.9% of patients were “quite or very” worried about the potential impairment of vital organs, 74.5% about increasing severity of symptoms, 68.9% about a possible onset of physical disability, and 63.3% about dealing with physical pain (Table 3). In addition, over half of the patients (52.8%) appeared troubled about the possibility of having to give up their job because of the disease; in particular, younger age groups were mainly concerned about not finding a job or having to give it up. Males seemed more concerned than women about the growing severity of symptoms (81.4% versus 69.8%), possible physical disability (74.4% versus 65.1%), impairment of vital organs (88.4% versus 81.0%), and also about any problems that could force them to give up their jobs (58.1% versus 49.2%). As expected, these negative feelings tended to worsen with age. Once again, male patients were the most worried about a future physical disability, particularly those aged 60 years or over.

## 6. Health-Related Quality of Life

Using the Scalone’s algorithm (for Italian population), the average HR-QoL of patients surveyed was equal to 0.72 QALY (Table 4). The average VAS score was 0.65 (0.66 for women and 0.63 for men). Comparing our results with those obtained by Scalone et al. [8] in a cohort of 6800 healthy individuals, we can confirm that Fabry patients have a poor QoL (0.72 versus 0.92 for healthy individuals).

The use of tariffs referring to the United Kingdom population allowed for drawing a comparison with data published in 2005 by Hoffmann et al. (Fabry Outcome Survey, FOS), resulting from a sample of 120 people (mix of predominantly European countries) with FD [17]. In that study, patients interviewed had an average HR-QoL of 0.66, without gender differences (0.66 for males and 0.67 for females), significantly lower than those reported for the general United Kingdom population; FOS results were comparable to our findings, obtained using the algorithm of Dolan et al. [16] (UK population): 0.68 QALY.

Moreover, the comparison with a previous cost-of-illness study enabled us to note that, on average, FD patients had a poorer QoL than people suffering from other chronic diseases, such as chronic hepatitis (0.90 QALY), cirrhosis (0.82 QALY), hepatocellular cirrhosis carcinoma (0.80 QALY) and orthotopic liver transplantation (0.83 QALY) [11]. The QoL of FD patients was also lower than in patients suffering from Crohn’s disease. In fact, according to a survey conducted in Italy from 2012 to 2013, which enrolled 552 patients with Crohn’s disease, the average value determined through the Italian algorithm was 0.76 [18]. According to our findings, FD patients had a slightly lower QoL than patients suffering from multiple sclerosis. In fact, according to a study published in 2017 [19], the latter showed a mean HR-QoL of 0.74.

## 7. Discussion and Conclusions

The World Health Organization (WHO) defines health as “a state of complete physical, mental and social well-being, and not simply as the absence of disease and infirmity”. Indeed, health cannot be considered only as a clinical outcome, and QoL has to be part of the measure to estimate a patient’s health status [20]. In particular, we have considered QoL, expressed through QALY, as the outcome par excellence in economic evaluations [21]. In general terms, all HR-QoL measures, such as QALYs, try to provide a complete evaluation of different treatments on different patients’ aspects, including emotional issues and how pain can impact the everyday life [22]. Children with lysosomal storage disorders have frequently lower HR-QoL across all domains relative to healthy counterparts [23]. Decreased physical functioning compared to healthy peers can often be encountered in many of these metabolic disorders, with implications in the affected organs, such as the skeletal system in Gaucher disease [24] or the heart and sense organs in the mucopolysaccharidoses [25,26].

FD is a lysosomal disease in which mutations affect the *GLA* gene located on the X chromosome: its defective product, the enzyme alpha-galactosidase A, causes accumulation of complex glycolipids, contributing to the disruption of cell function in several organs, including kidneys, heart, nervous system and skin, with variable severity and consequent damage of organ function. In addition to the debilitating physical symptoms, there are also under-recognized and poorly characterized psychiatric features in FD. Different studies have shown that FD registries improved our understanding of patients’ real-world experience also in relation to enzyme replacement [4,27,28]. Phenotypic heterogeneity of FD has hindered the monitoring of QoL assessment in different cohorts of patients, though female heterozygotes appear more studied than male hemizygotes. In general terms, the QoL is globally reduced due to fatigue, exercise intolerance and poor self-perception of health. An evaluation conducted in Germany showed that different organ involvement in female patients with Fabry disease can impact QoL with different significance [29].

To the best of our knowledge, our survey is the largest ever conducted on Italian patients with FD to assess their QoL, having enrolled 106 patients.

Our study confirmed that patients with FD have a worse average QoL than the general population, but also lower than in patients with other chronic diseases, such as Crohn’s disease, hepatitis, cirrhosis and, albeit slightly, patients with multiple sclerosis. Our results confirmed that FD subverts many aspects of patients’ lives, influencing their perception of themselves, state of mind, work and social choices. Moreover, we found anxiety, depression, loneliness and feeling of rejection as important indicators of decreased QoL. In particular, we found a slight prevalence of depression in females, differently from what was reported in an Australian study [12]. Moreover, FD patients showed QALY values lower than in other chronic diseases, such as cirrhosis or multiple sclerosis [30,31]. The causes of this psychological distress are multiple, including awareness of an inexorable progressive and non-reversible pathology.

The results of this survey lead to several considerations on the management of FD. Firstly, the study highlights the importance of building a psychological support network for FD patients, with the aim of facilitating the acceptance of their condition and its emotional burden. The many reference centers for FD should be equipped with specialized professionals in this field and should work alongside multidisciplinary teams that assess patients in their care pathway. In addition, the importance of patients’ associations should not be underestimated, providing opportunities for meetings and support. A further point that could improve the overall perception of FD is disease knowledge in the general population, including medical professionals. In fact, it is not uncommon to observe patients who are diagnosed after years of many medical examinations and investigations: this can generate distrust and a feeling of not being understood, but, most of all, causes diagnostic delay with disease progression and significant decrease in treatment efficacy or in life expectancy. An implementation of multidisciplinary psychosocial interventions may be particularly beneficial in adolescent patients to prevent poor QoL and psychosocial functioning in adulthood.

In conclusion, our study showed that Italian FD patients have a poor average QoL, and that it is also lower if compared with patients presenting other inflammatory chronic disorders, such as Crohn’s disease, chronic hepatitis, cirrhosis and multiple sclerosis. The findings further confirm that FD affects patients’ lives and influences their social choices. The feeling of rejection as well as the sensitivity to other people’s opinions are among the psychological motivations that characterize the most vulnerable patients. Combined with the awareness of an impossible recovery, this often causes mood disorders which might further worsen the overall impact of the disease.

## Figures and Tables

**Figure 1 diseases-09-00072-f001:**
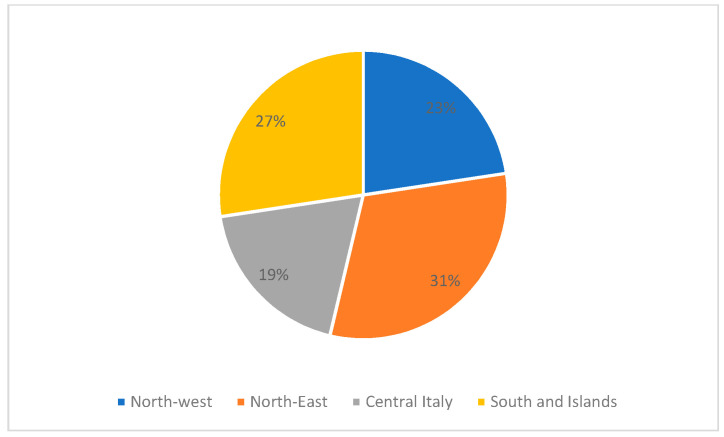
Geographical distribution of Italian patients with Fabry disease in our study.

**Table 1 diseases-09-00072-t001:** General characteristics of patients with Fabry disease considered in the study.

Variable	M + F	M	F
Sample	106	43	63
Average age (years)	42	37	45
Minimum age (years)	5	7	5
Maximum age (years)	77	67	77
Average age at diagnosis (years)	32	27	35
% of females	59.5%		
**Status**			
Married	54.7%	44.2%	61.9%
Unmarried	33.0%	48.8%	22.2%
Divorced	8.5%	7.0%	9.5%
Separated	3.8%	0.0%	6.3%
Widow/widower	54.7%	44.2%	61.9%
**School Level and Qualifications**			
Degree	20.8%	18.6%	22.2%
High school diploma	44.3%	48.8%	41.3%
2–3-year diploma	5.7%	4.7%	6.3%
Middle school diploma	20.8%	18.6%	22.2%
Primary school diploma	6.6%	77.0%	6.3%
No diploma or degree	1.9%	2.3%	1.6%
**Professional Status**			
Employed	53.8%	60.5%	49.2%
In search of his/her first job	0.0%	0.0%	0.0%
Unemployed	9.4%	11.6%	7.9%
Retired	8.5%	4.7	11.1
Student	11.3%	14.0	9.5
Housewife	9.4%	0.0%	15.9%
Unable to work	1.9%	4.7%	0.0%
Other	5.7%	4.7%	6.3%

**Table 2 diseases-09-00072-t002:** Results obtained with the EQ5D questionnaire in our cohort of patients with Fabry disease.

Mobility	
I have no problems in walking about.	70.75%
I have some problems in walking about.	28.30%
I am confined to bed.	0.94%
Self-Care	
I have no problems with self-care.	91.51%
I have some problems washing or dressing myself.	7.55%
I am unable to wash or dress myself.	0.94%
Usual Activities	
I have no problems with performing my usual activities.	52.83%
I have some problems with performing my usual activities.	45.28%
I am unable to perform my usual activities.	1.89%
Pain/Discomfort	
I have no pain or discomfort.	22.64%
I have moderate pain or discomfort.	68.87%
I have extreme pain or discomfort.	8.49%
Anxiety/Depression	
I am not anxious or depressed.	42.45%
I am moderately anxious or depressed.	50.94%
I am extremely anxious or depressed.	6.60%

**Table 3 diseases-09-00072-t003:** Findings resulting from the evaluation of the questionnaire in our cohort of patients with Fabry disease.

**The Psychological Impact of the Disease (% of Patients)**												
	**Loneliness and Isolation**				**Decision to Have Children**				**Anxiety**			
	Not at all	Little	Quite	Much	Not at all	Little	Quite	Much	Not at all	Little	Quite	Much
Total	45.3	33.0	17.9	3.8	44.3	15.1	17.9	22.6	21.7	32.1	33.0	13.2
**Gender ***												
Male	51.2	30.2	16.3	2.3	39.5	16.3	23.3	20.9	27.9	32.6	32.6	7.0
Female	41.3	34.9	19.0	4.8	47.6	14.3	14.3	23.8	17.5	31.7	33.3	17.5
**Age (years)**												
18–29	55.6	33.3	11.1	0.0	11.1	11.1	55.6	22.2	22.2	11.1	55.6	11.1
30–39	45.5	40.9	13.6	0.0	9.1	27.3	27.3	36.4	22.7	31.8	31.8	13.6
40–49	44.0	32.0	20.0	4.0	48.0	12.0	12.0	28.0	16.0	36.0	24.0	24.0
50–59	40.9	36.4	18.2	4.5	50.0	27.3	4.5	18.2	9.1	45.5	45.5	0.0
60–69	25.0	41.7	25.0	8.3	83.3	0.0	0.0	16.7	25.0	25.0	33.3	16.7
70+	0.0	0.0	100.0	0.0	66.7	0.0	33.3	0.0	33.3	0.0	66.7	0.0
	**Depression**				**Relational Problems**				**Fear of Other People’s Judgement**			
	Not at all	Little	Quite	Much	Not at all	Few	Quite	Many	Not at all	Little	Quite	Much
Total	37.7	39.6	18.9	3.8	57.5	30.2	7.5	4.7	50.9	24.5	18.9	5.7
**Gender**												
Male	39.5	39.5	18.6	2.3	53.5	30.2	9.3	7.0	41.9	25.6	20.9	11.6
Female	36.5	39.7	19.0	4.8	60.3	30.2	6.3	3.2	57.1	23.8	17.5	1.6
**Age (years)**												
18–29	44.4	44.4	11.1	0.0	55.6	44.4	0.0	0.0	33.3	44.4	11.1	11.1
30–39	36.4	45.5	13.6	4.5	63.6	31.8	4.5	0.0	45.5	31.8	18.2	4.5
40–49	32.0	44.0	20.0	4.0	48.0	40.0	8.0	4.0	52.0	28.0	16.0	4.0
50–59	18.2	54.5	27.3	0.0	63.6	13.6	13.6	9.1	68.2	9.1	18.2	4.5
60–69	50.0	25.0	16.7	8.3	50.0	41.7	0.0	8.3	50.0	25.0	25.0	0.0
70+	33.3	0.0	66.7	0.0	33.3	0.0	66.7	0.0	33.3	0.0	66.7	0.0
**Disease-related Problems (% Patients)**												
	**Chronic/Persistent Pain**				**Acute/Temporary Pain**				**Fatigue**			
	Not at all	Little	Quite	Much	Not at all	Little	Quite	Much	Not at all	Little	Quite	Much
Total	27.4	33.0	30.2	9.4	20.8	32.1	38.7	8.5	7.5	29.2	40.6	22.6
**Gender**												
Male	27.9	32.6	25.6	14.0	11.6	32.6	41.9	14.0	14.0	27.9	37.2	20.9
Female	27.0	33.3	33.3	6.3	27.0	31.7	36.5	4.8	3.2	30.2	42.9	23.8
**Age (years)**												
18–29	33.3	44.4	11.1	11.1	33.3	11.1	44.4	11.1	22.2	22.2	33.3	22.2
30–39	36.4	27.3	22.7	13.6	27.3	18.2	50.0	4.5	4.5	45.5	31.8	18.2
40–49	12.0	28.0	44.0	16.0	8.0	48.0	28.0	16.0	0.0	24.0	56.0	20.0
50–59	22.7	45.5	31.8	0.0	22.7	31.8	40.9	4.5	4.5	13.6	45.5	36.4
60–69	25.0	25.0	41.7	8.3	25.0	33.3	33.3	8.3	8.3	16.7	50.0	25.0
70+	0.0	33.3	66.7	0.0	33.3	33.3	33.3	0.0	0.0	0.0	66.7	33.3
	**Motor Difficulties**											
	Not at all	Few	Quite	Many								
Total	40.6	32.1	20.8	6.6								
**Gender**												
Male	37.2	32.6	23.3	7.0								
Female	42.9	31.7	19.0	6.3								
**Age (years)**												
18–29	55.6	33.3	11.1	0.0								
30–39	50.0	31.8	13.6	4.5								
40–49	28.0	40.0	32.0	0.0								
50–59	36.4	22.7	31.8	9.1								
60–69	41.7	33.3	8.3	16.7								
70+	0.0	0.0	66.7	33.3								
**Concerns About the Future (% Patients)**												
	**Increasing Severity of Symptoms**				**Physical Disability**				**Physical Pain**			
	Not at all	Little	Quite	Much	Not at all	Little	Quite	Much	Not at all	Little	Quite	Much
Total	5.7	19.8	50.0	24.5	12.3	18.9	45.3	23.6	6.6	30.2	42.5	20.8
**Gender**												
Male	4.7	14.0	53.5	27.9	14.0	11.6	48.8	25.6	7.0	30.2	39.5	23.3
Female	6.3	23.8	47.6	22.2	11.1	23.8	42.9	22.2	6.3	30.2	44.4	19.0
**Age (years)**												
18–29	11.1	11.1	44.4	33.3	33.3	11.1	44.4	11.1	11.1	22.2	44.4	22.2
30–39	4.5	22.7	40.9	31.8	4.5	27.3	40.9	27.3	0.0	40.9	31.8	27.3
40–49	4.0	24.0	64.0	8.0	8.0	20.0	52.0	20.0	8.0	24.0	60.0	8.0
50–59	4.5	18.2	50.0	27.3	13.6	18.2	40.9	27.3	4.5	36.4	31.8	27.3
60–69	8.3	16.7	41.7	33.3	8.3	16.7	50.0	25.0	16.7	8.3	50.0	25.0
70+	0.0	0.0	66.7	33.3	0.0	0.0	66.7	33.3	0.,0	0.0	66.7	33.3
	**Impairment of Vital Organs**				**Possibility of Having to Give up/not Finding a Job**							
	Not at all	Little	Quite	Much	Not at all	Little	Quite	Much				
Total	4.7	11.3	48.1	35.8	24.5	22.6	30.2	22.6				
**Gender**												
Male	7.0	4.7	41.9	46.5	16.3	25.6	18.6	39.5				
Female	3.2	15.9	52.4	28.6	30.2	20.6	38.1	11.1				
**Age (years)**												
18–29	11.1	0.0	33.3	55.6	11.1	33.3	22.2	33.3				
30–39	0.0	4.5	45.5	50.0	0.0	31.8	36.4	31.8				
40–49	0.0	12.0	64.0	24.0	16.0	20.0	44.0	20.0				
50–59	0.0	27.3	45.5	27.3	31.8	27.3	22.7	18.2				
60–69	8.3	8.3	50.0	33.3	66.7	8.3	16.7	8.3				
70+	0.0	0.0	66.7	33.3	66.7	0.0	33.3	0.0				

* Chi squared for gender (** *p* ≤ 0.1): Loneliness and isolation *p* = 0.7; Decision to have children *p* = 0.6; Anxiety *p* = 0.3; Depression *p* > 0.9; Relational problems *p* = 0.7; Fear of other people’s judgement *p* = 0.1 **; Chronic/persistent pain *p* = 0.6; Acute/temporary pain *p* = 0.1 **; Fatigue *p* = 0.2; Motor difficulties *p* > 0.1; Increasing severity of symptoms *p* = 0.6; Physical disability *p* = 0.4; Physical pain *p* = 0.5; Impairment of vital organs *p* = 0.1 **; Possibility of having to give up/not finding a job *p* = 0.0 **.

**Table 4 diseases-09-00072-t004:** Results of the health-related quality of life (HR-QoL) scores in our cohort of patients with Fabry disease.

	Italian Tariffs	UK Tariffs
Total	0.72	0.68
**Gender**		
Male	0.72	0.68
Female	0.71	0.68
**Age**		
18–29	0.86	0.77
30–39	0.82	0.74
40–49	0.79	0.63
50–59	0.77	0.64
60–69	0.64	0.64
70+	0.69	0.40

UK: United Kingdom.

## Data Availability

The data that support the findings of this study are available from the corresponding author on specific request.

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
