# Peer review of "Survey about the Quality of Life of Italian Patients with Fabry Disease"

_diseases, 2021, doi:10.3390/diseases9040072_

Round 1

Reviewer 1 Report

I would like to see this manuscript published; however revisions are necessary.  

anuscript review for Diseases

“Survey about the Quality of Life of Italian Patients with Fabry Disease”

Overall

The authors may benefit from having a native English speaker proof-read the manuscript, if at all possible, as there are many instances where the wrong word is used in a sentence.

In addition, analytical statistics need to be performed on the data to provide proof that the comparisons made in this manuscript were not obtained by chance.  The sample size (106 subjects) is sufficient to provide more than simply descriptive statistics in Table 3 and the Results section.

Finally, inclusion of more recent references / citations in both the Introduction and Discussion will be helpful (see individual comments below).

Abstract

Line 34 states that “Our study upholds that FD patients have a poorer OoL.”  Two revisions requested – 1) Change OoL to QoL and 2) poorer QoL than whom? (provide a comparison group in this sentence).

Introduction

Page 2, lines - Please add at least a few more recent references for the occurrence of depression and neuropsychological deficits in individuals with Fabry.  It is ok to have more than one citation in parentheses, in order to include both classic articles and more recent evidence.

Page 2, line 69 – Please change the beginning of the sentence from “Aim of this study…” to “The aim of this study….” for grammatical reasons.

Methods

Page 2, line 76 – I am confused by the use of the word ‘autonomously’ in connection with answering the survey.  Can you please clarify the meaning here?

Page 2, line 81 – I don’t think ‘synthetic’ is the word you mean to use here, as in English this means ‘a substance made by chemical synthesis, especially to imitate a natural product, such as a textile fiber.”

Results

Page 3, line 102 – What do you mean by the phrase, “The survey involved 162 patients, 106 of whom completed the questionnaire”?  If only 106 completed the survey, did you mean to imply you used partial data from the rest?  Or did you mean to suggest that 162 were sent the survey but only 106 responded by completing the survey?  Please reword this sentence for greater clarity.

Page 3, line 105 – Please remove the word ‘the’ before ’59.5%’ for grammatical reasons.

Page 4, line 113 – Please replace the phrase ‘the specific treatment’ with “Fabry-specific treatment”

Table 2 – The section under ‘Usual Activites’ talks about pain rather than activities.  Please correct.

Table 3 would be enhanced greatly by analytic statistics with p-values, to explore whether the differences seen between categories in the table are statistically significant or not.  With 106 subjects, comparative analysis should not be limited to descriptive statistics.  This will enable you to rewrite and improve your results section, and will increase the valuable contribution of your paper on Fabry.

Page 7, line 135 – please rephrase ‘conditioning of the plan to have children’ to ‘the impact of Fabry on reproductive decision-making’ for clarity and grammatical reasons.

Page 7, lines 139-141 – this is a great example of why analytic statistics are needed here.   You cannot say ‘our data confirmed that problems were more relevant for women” without providing a significant p-value for this comparison to prove the difference was not obtained merely by chance.178-

Page 8, lines 180 – Again, you cannot use the words ‘significant difference’ in research without statistics proving the difference is significant.

Page 8, lines 184-193 – I love the use of information on other chronic diseases, rather than just comparison with healthy controls.  Were the studies with other chronic diseases also conducted on Italian patients or were they conducted with other populations?

Discussion

As with the Introduction, please add at least a few more recent references for comparative studies including QOL measures in individuals with Fabry.  While these may not have been done in Italy before or been the primary focus of research in other studies, QOL in Fabry disease has certainly been examined before or included as a secondary measure in other research internationally.

Page 9, line 221 – Please do analytical statistics to confirm this is not by chance.  Can also cite other studies showing equal rates of depression in men and women with Fabry disease, unlike patterns in healthy populations.

Page 9 – lines 226-240 – Excellent recommendations!!

Page 9 – lines 236-237 – Please include citations of previous research documenting diagnostic delay of Fabry disease, leading to treatment delay and disease progression.  Some articles even give an average range of years after symptom onset until diagnosis of Fabry, which would be a good citation here.

Author Response

REVIEWER 1

Abstract

Line 34 states that “Our study upholds that FD patients have a poorer OoL.”  Two revisions requested – 1) Change OoL to QoL and 2) poorer QoL than whom? (provide a comparison group in this sentence).

Received and done

Introduction

Page 2, lines - Please add at least a few more recent references for the occurrence of depression and neuropsychological deficits in individuals with Fabry.  It is ok to have more than one citation in parentheses, in order to include both classic articles and more recent evidence.

Received (after the 5th reference we have added some newer ones)

Page 2, line 69 – Please change the beginning of the sentence from “Aim of this study…” to “The aim of this study….” for grammatical reasons.

Received

Methods

Page 2, line 76 – I am confused by the use of the word ‘autonomously’ in connection with answering the survey.  Can you please clarify the meaning here?

Received

Page 2, line 81 – I don’t think ‘synthetic’ is the word you mean to use here, as in English this means ‘a substance made by chemical synthesis, especially to imitate a natural product, such as a textile fiber.”

Received

Results

Page 3, line 102 – What do you mean by the phrase, “The survey involved 162 patients, 106 of whom completed the questionnaire”?  If only 106 completed the survey, did you mean to imply you used partial data from the rest?  Or did you mean to suggest that 162 were sent the survey but only 106 responded by completing the survey?  Please reword this sentence for greater clarity.

106, out of the 162 patients who started filling out the survey, completed it. The analysis were carried out on 106 patients

Page 3, line 105 – Please remove the word ‘the’ before ’59.5%’ for grammatical reasons.

Received

Page 4, line 113 - Please replace the phrase ‘the specific treatment’ with “Fabry-specific treatment”

Received

Table 2 - The section under ‘Usual Activites’ talks about pain rather than activities.  Please correct.

Received

Table 3 would be enhanced greatly by analytic statistics with p-values, to explore whether the differences seen between categories in the table are statistically significant or not.  With 106 subjects, comparative analysis should not be limited to descriptive statistics.  This will enable you to rewrite and improve your results section, and will increase the valuable contribution of your paper on Fabry.

We added the p-value for gender; see Table 3

Page 7, line 135 – please rephrase ‘conditioning of the plan to have children’ to ‘the impact of Fabry on reproductive decision-making’ for clarity and grammatical reasons.

Received (and corrected the sentence)

Page 7, lines 139-141 – this is a great example of why analytic statistics are needed here.   You cannot say ‘our data confirmed that problems were more relevant for women” without providing a significant p-value for this comparison to prove the difference was not obtained merely by chance.178-

Correct: we added the p-value

Page 8, lines 180 – Again, you cannot use the words ‘significant difference’ in research without statistics proving the difference is significant.

Idem

Page 8, lines 184-193 – I love the use of information on other chronic diseases, rather than just comparison with healthy controls.  Were the studies with other chronic diseases also conducted on Italian patients or were they conducted with other populations?

Received (information were inserted)

Discussion

As with the Introduction, please add at least a few more recent references for comparative studies including QOL measures in individuals with Fabry.  While these may not have been done in Italy before or been the primary focus of research in other studies, QOL in Fabry disease has certainly been examined before or included as a secondary measure in other research internationally.

The references from #5 to #9 were firstly added. Some newer references, as required, have been at page 9.

Page 9, line 221 – Please do analytical statistics to confirm this is not by chance.  Can also cite other studies showing equal rates of depression in men and women with Fabry disease, unlike patterns in healthy populations.

The cited studies show different rates of psychiatric issues which might differ according to several geographical and social aspects.

Page 9 – lines 226-240 – Excellent recommendations!!

Thanks

Page 9 – lines 236-237 – Please include citations of previous research documenting diagnostic delay of Fabry disease, leading to treatment delay and disease progression.  Some articles even give an average range of years after symptom onset until diagnosis of Fabry, which would be a good citation here.

The reference #30 shows how a diagnostic delay drags both treatment delay and overt occurrence of complications.

Reviewer 2 Report

The authors present a very important problem of patients with Fabry disease (FD)- their poor quality of life (QoL). The multiorgan damage and many symptoms result in social and economic exclusion in many cases.
Data were collected from a cohort of Italian FD patients using an innovative and progressive method Computer-Assisted Personal Interview (CAPI). The questionnaire was detailed and included many functional and psychological aspects of health-related QoL.
The study group was very large and representative in terms of age and gender. The authors reveal that QoL FD patients remain extremely poor despite applied enzyme replacement therapy (ERT) and worse than in some other chronic inflammatory diseases. Significant differences in QoL were
found between younger and older patients and males and females. Moreover, this observational survey indicates that FD patients need not only clinical but also long live psychological support.

Author Response

The authors present a very important problem of patients with Fabry disease (FD)- their poor quality of life (QoL). The multiorgan damage and many symptoms result in social and economic exclusion in many cases.
Data were collected from a cohort of Italian FD patients using an innovative and progressive method Computer-Assisted Personal Interview (CAPI). The questionnaire was detailed and included many functional and psychological aspects of health-related QoL. The study group was very large and representative in terms of age and gender. The authors reveal that QoL FD patients remain extremely poor despite applied enzyme replacement therapy (ERT) and worse than in some other chronic inflammatory diseases. Significant differences in QoL were found between younger and older patients and males and females. Moreover, this observational survey indicates that FD patients need not only clinical but also long live psychological support.

Thanks for your comment

Reviewer 3 Report

The paper by Spandonaro et al described the results of a survey performed in a cohort of Italian people suffering from Fabry´s Disease.

The authors should receive credits for all their efforts. However, there are several shortcomings regarding the paper. The whole manuscript is entirely descriptive and regarding survey two aspects are striking:

     First: the relatively low rate of responders

  Second: the predominance of females taking into account that Fabry Disease is X-linked and probably clinical phenotype in males is more severe.

These two aspects could be an important source of bias and could interfere with the interpretation of the results.

There is no mention of the relationship between  main clinical phenotype and different aspects of QoL or what are the main determinants of poorer QoL

Discussion and Conclusion section rows 207 to 209: This paragraph is redundant, description of Fabry's Disease has been mentioned in the introduction section.

Both, in the abstract and in the conclusions, it is stated that the QoL is poorer in Fabry's Diseases than in other chronic diseases such as Crohn's Diseases, Multiple Sclerosis. However, the authors do not provide direct data or references to support this statement.

Author Response

The paper by Spandonaro et al described the results of a survey performed in a cohort of Italian people suffering from Fabry´s Disease.

The authors should receive credits for all their efforts. However, there are several shortcomings regarding the paper. The whole manuscript is entirely descriptive and regarding survey two aspects are striking:

First: the relatively low rate of responders 

Of course, the response rate was lower than expected, highlighting the difficulty in conducting epidemiologic studies in patients affected by chronic non-cancer diseases. Many clinical trial groups have frequently reported on the challenges of implementing QOL research in metabolic diseases characterized by progressive course and sometimes fatal destiny, like Fabry disease. 

Second: the predominance of females taking into account that Fabry disease is X-linked and probably clinical phenotype in males is more severe. These two aspects could be an important source of bias and could interfere with the interpretation of the results. 

The fact that women represented 59.5% of our sample (57.1% said they felt “moderately” and 7.9 “extremely” anxious or depressed, compared to 41.9% and 4.7% of males) matches with the fact that females heterozygous for mutations in the α-galactosidase A gene are less severely affected and more worry to improve the overall quality of their lives. In past times female carriers were considered to be largely asymptomatic. Only more recently our knowledge on Fabry disease has revealed that Fabry females experience significant clinical manifestations involving the central nervous system and heart, similar to those seen in male hemizygotes. A potential limitation of our study is related to ascertainment bias, due to enrollment of patients on a voluntary basis, independently from the overall disease burden and response to therapy. We can also hypothesize that the shortage of male  patients may be due to greater disease severity in male hemizygotes, where the disease progresses and worsens over time, have a shorter survival than 10 years and are especially reluctant to participate in observational studies. 

There is no mention of the relationship between  main clinical phenotype and different aspects of QoL or what are the main determinants of poorer QoL 

Phenotypic heterogeneity of Fabry disease has hindered the monitoring of QoL assessment in different cohorts of patients, though female heterozygotes appear more studied than male hemizygotes. In general terms, the QoL is globally reduced due to fatigue, exercise intolerance and poor self-perception of health. An evaluation conducted in Germany showed that different organ involvement in female patients with Fabry disease can impact on QoL with different significance.  This has been inserted in the Discussion and a new reference (#31) has been added.

Discussion and Conclusion section rows 207 to 209: This paragraph is redundant, description of Fabry's Disease has been mentioned in the introduction section.

Thanks for the opportunity of clarification. The short redefinition of the disease is simply the backstory to introduce the results of our survey.

Both, in the abstract and in the conclusions, it is stated that the QoL is poorer in Fabry's Diseases than in other chronic diseases such as Crohn's Diseases, Multiple Sclerosis. However, the authors do not provide direct data or references to support this statement.

Received and done

Reviewer 4 Report

The manuscript entitled Survey about the Quality of Life of Italian Patients with Fabry Disease describes the QoL of 106 Italian FD patients.

The present manuscript provides interesting results but need to be presented in a more comprehensive way. Please give response to the following comments:

What is the main contribution of this work? The characteristics of QoL of FD patients, in general, have already been described in different publications.

Was the Italian adaptation of the questionnaire validated?

With respect to the Table 1 (General characteristics of patients with Fabry disease considered in the study) it would be better to separate the percentages by sex.

With respect to the Table 3 (Findings resulting from the evaluation of the questionnaire in our cohort of patients with Fabry disease) there are too many data and is difficult to read. To facilitate an easy comprehension by the reader I recommend to group different ages for the presentation (for example 18-49, 50+, 70+) in accord with the results. The complete data of all the questionnaire may be included in an Appendix in Supplemental Materials.

The results of table 4 are not clear. Please, clarify the comparison of HRQoL in the present study with the FOS study, and with the Italian population reference and UK tariffs. Please, state that the FOS cohort in the 2005 paper is a mix of predominantly European countries (Italy also among them).

Author Response

What is the main contribution of this work? The characteristics of QoL of FD patients, in general, have already been described in different publications.

To the best of our knowledge, the present survey (promoted by the Anderson-Fabry Italian Association), performed through an adaptation made by the EuroQoL Group, is the largest one to date carried out on Italian patients suffering from FD through a web-based platform. The survey has compared the results obtained with data related to other chronic disabling diseases

Was the Italian adaptation of the questionnaire validated?

Yes, the 5D EQ has also been validated for Italy

With respect to the Table 1 (General characteristics of patients with Fabry disease considered in the study) it would be better to separate the percentages by sex.

Received and done

With respect to the Table 3 (Findings resulting from the evaluation of the questionnaire in our cohort of patients with Fabry disease) there are too many data and is difficult to read. To facilitate an easy comprehension by the reader I recommend to group different ages for the presentation (for example 18-49, 50+, 70+) in accord with the results. The complete data of all the questionnaire may be included in an Appendix in Supplemental Materials.

It seemed important to us to show the differences by age of the population, but if preferred we could insert these data in an Appendix. Of course

The results of Table 4 are not clear. Please, clarify the comparison of HRQoL in the present study with the FOS study, and with the Italian population reference and UK tariffs. Please, state that the FOS cohort in the 2005 paper is a mix of predominantly European countries (Italy also among them).

Received: the sentence has been modified

Round 2

Reviewer 1 Report

The authors have done a great job revising the manuscript.  Only a few changes remain to be made!

Please add a data analysis section to your methods.  Right now, the manuscript says descriptive analysis was carried out, but in the revisions you have appropriatesly added values from a Chi squared analysis.  This needs to be outlined in your methods so that others can replicate if they need. 

Also in this section, please include the designated level at which p-values are determined to be significant.  This is important for interpreting your findings.

Please double check the p-values listed for Table 3.  Did you mean to state that Depression p>0.9 or p<0.09?  It is difficult to tell from the direction of the <>.  Ditto for motor difficulties. 

Finally, please place an asterisk next to the p-values in Table 3 that meet your designated level of significance.

Author Response

The authors have done a great job revising the manuscript.  Only a few changes remain to be made!

Thanks

Please add a data analysis section to your methods.  Right now, the manuscript says descriptive analysis was carried out, but in the revisions you have appropriatesly added values from a Chi squared analysis.  This needs to be outlined in your methods so that others can replicate if they need. 

Added, thanks

Also in this section, please include the designated level at which p-values are determined to be significant.  This is important for interpreting your findings.

Added, thanks

Please double check the p-values listed for Table 3.  Did you mean to state that Depression p>0.9 or p<0.09?  It is difficult to tell from the direction of the <>.  Ditto for motor difficulties. 

p>0,9 is correct

Finally, please place an asterisk next to the p-values in Table 3 that meet your designated level of significance.

Added, thanks

Reviewer 3 Report

The overall quality of the paper has been improved and the authors had answered the comments.

No further comments

Author Response

Thank you very much